# Tunable transmission of quantum Hall edge channels with full degeneracy lifting in split-gated graphene devices

Katrin Zimmermann[1,2], Anna Jordan[1,2], Frédéric Gay[1,2], Kenji Watanabe[3], Takashi Taniguchi[3], Zheng Han[1,2], Vincent Bouchiat[1,2], Hermann Sellier[1,2] & Benjamin Sacépé[1,2]

Charge carriers in the quantum Hall regime propagate via one-dimensional conducting channels that form along the edges of a two-dimensional electron gas. Controlling their transmission through a gate-tunable constriction, also called quantum point contact, is fundamental for many coherent transport experiments. However, in graphene, tailoring a constriction with electrostatic gates remains challenging due to the formation of p–n junctions below gate electrodes along which electron and hole edge channels co-propagate and mix, short circuiting the constriction. Here we show that this electron–hole mixing is drastically reduced in high-mobility graphene van der Waals heterostructures thanks to the full degeneracy lifting of the Landau levels, enabling quantum point contact operation with full channel pinch-off. We demonstrate gate-tunable selective transmission of integer and fractional quantum Hall edge channels through the quantum point contact. This gate control of edge channels opens the door to quantum Hall interferometry and electron quantum optics experiments in the integer and fractional quantum Hall regimes of graphene.

[1] Univ. Grenoble Alpes, Institut Néel, F-38000 Grenoble, France. [2] CNRS, Institut Néel, F-38000 Grenoble, France. [3] Advanced Materials Laboratory, National, Institute for Materials Science, 1-1 Namiki, Tsukuba 305, Japan. Correspondence and requests for materials should be addressed to B.S. (email: benjamin.sacepe@neel.cnrs.fr).

In two-dimensional electron gases formed in semiconductor heterostructures, confinement of electron transport through nano-patterned constrictions has led to tremendous advances in quantum transport experiments[1]. Chief among key devices, operating both at zero magnetic field and in the quantum Hall (QH) regime under strong magnetic field, is the quantum point contact (QPC): a gate-defined narrow and short constriction that enables control over the exact number of transmitted electronic modes between two reservoirs of electrons, leading to conductance quantization[2–7]. In the QH regime, fine-tuning of transmission across the QPC via electrostatic gating has become essential for many experiments based on electron tunnelling and charge partitioning, such as shot noise measurements[8,9], QH interferometry[10,11] and electron quantum optics[12,13].

Yet, in monolayer graphene, demonstration of QPC operation in split-gate geometry remains challenging. The major hurdle precluding engineering split-gated constrictions stems from the gapless graphene electronic band structure[14]. Depletion of an electron-doped region with a gate electrode indeed leads to a hole-doped region, creating a conducting, gapless p–n junction that inevitably short circuits the constriction.

The alternative route that consists in confining electron transport through etched constrictions has long been difficult due to the fact that physically etched constrictions in low-mobility devices are subject to electron localization by disorder and charging effects[15–18]. Recent improvements in device fabrication techniques have solved these issues with a significant rise of graphene mobility, mitigating disorder effects. In turn, remarkable etched constrictions in suspended graphene flakes were realized, showing clear conductance quantization upon varying the global device charge carrier density at zero magnetic field[19], and later confirmed in encapsulated graphene devices[20].

In the QH regime, tunability of the QPC constriction with split-gate electrodes is mandatory to control the transmission of QH edge channels and the tunnelling between counter-propagating QH edge channels. However, the p–n junction formed along gate electrodes also poses problems as electron-type QH edge channels co-propagate along the junction with hole-type edge channels, as illustrated in Fig. 1a. First experiments in devices equipped with a single top gate have shown that disorder promotes charge transfer between these co-propagating electron and hole edge channels, leading to chemical potential equilibration[21–23]. Recently, it has been shown that the use of boron nitride (hBN) substrates that considerably reduce disorder can suppress equilibration effects at the p–n interface in single top-gate devices[24]. The origin of this suppression remains unclear and could result from the opening of a gap at the charge neutrality point as observed in some graphene-on-hBN devices[25,26]. For the split-gate defined QPC geometry, experiments in graphene QPC devices operating on the fourfold degenerate Landau levels (LLs) showed that the presence of QH edge channel mixing is also detrimental as it creates a short circuit of the constriction via localized channels beneath the split gates (red channels in Fig. 1a), thus hindering gate control of QH edge channel transmission through the QPC[27,28].

In this work we employ high-mobility hBN/graphene/hBN van der Waals heterostructures[29,30] to take advantage of the full symmetry breaking of the LLs and the emergence of an energy gap between electron and hole LLs that significantly mitigate QH edge channel mixing. The heterostructures are equipped with back-gate and split-gate electrodes to realize QPC devices operating in the QH regime. By continuously changing the graphene electron densities in the bulk and beneath the split-gates, we identify the exact edge channel configurations for which QH edge channels are immune to short circuiting. This enables to selectively gate-tune the transmission of both integer and fractional QH edge channels through the QPC, eventually leading to full pinch-off.

## Results

**Split-gated high-mobility graphene devices.** High-mobility samples were fabricated following the recent tour de force in graphene device fabrication techniques using van der Waals pick-up[30] that produces remarkably clean encapsulation of graphene in between two hBN flakes (see Supplementary Information for details). In this configuration a top hBN layer serves naturally as a high-quality dielectric for gating. Suitable etching of the hBN/graphene/hBN structure enables deposition of both contact electrodes on the edge of the hetero-structure and split-gate electrodes in a single metal deposition step.

In this study two different devices were fabricated, each showing quantitatively identical behaviours (see Supplementary Information). We present here the results of an hBN (17 nm)/graphene/hBN (32 nm) structure patterned in a 2 μm wide Hall bar with 6 contacts, and split gates of 150 nm gap located in the central part of the device (see Fig. 1a,b). The $SiO_2$(285 nm)/Si++ substrate serves as a back gate. The six contacts enable measurement of three voltages (see Fig. 1b) in four-terminal configurations, leading to the longitudinal and diagonal resistances $R_L$ and $R_D$, and the Hall resistance $R_H$. All measurements were carried out at a temperature of 0.05 K.

Figure 1c shows a map of $R_L$ versus back-gate and split-gate voltages, $V_{bg}$ and $V_{sg}$ respectively, at zero magnetic field. The charge neutrality point (CNP) of the bulk graphene—the resistance peak independent of $V_{sg}$—is located at $V_{bg}^{CNP} = -1$ V, indicating very small residual doping. The diagonal line drawn by a second peak in $R_L$ indicates charge neutrality in the split-gated region of graphene. Its slope corresponds to the ratio of capacitances between back-gated and split-gated regions ($C_{sg}/C_{bg} = 7$) and hence provides a way to assess quantitatively for the charge carrier density beneath the split gates (using $C_{bg} = 11.1$ nF cm$^{-2}$).

Analysis of transport properties at $V_{sg} = 0$ V leads to a mean free path of 1.8 μm that coincides with the width of the Hall bar, and a bulk mobility superior to 200 000 cm$^2$ V$^{-1}$ s$^{-1}$ at a charge carrier density of $n \sim 10^{12}$ cm$^{-2}$. These, together with signatures of negative nonlocal resistance (Supplementary Fig. 1), demonstrate ballistic transport in the device.

The quality of our device is also apparent in the resolution of broken symmetry states of graphene in the QH regime at moderate magnetic field ($B$). Figure 1d shows a colour map of $R_L$ versus $V_{bg}$ and $B$ taken at $V_{sg} = 0$ V (see Supplementary Fig. 2 for hole doping). In this Landau fan diagram, $R_L = 0$ blue strips indexed by their respective integer value of the bulk filling factor $\nu_b = n\phi_0/B$ ($n$ is the carrier density and $\phi_0 = h/e$ the flux quantum with $e$ the electron charge and $h$ the Planck constant) signal the presence of QH states. In addition to the usual graphene sequence $\nu_b = 4(N+\frac{1}{2}) = 2$, 6, 10... where $N$ is the LL index, broken symmetry states at $\nu_b = 1$ and at half-filling $\nu_b = 4$, 8, 12, are visible at fields as low as $B = 3$ T. At $B > 5$ T, LL degeneracies are fully lifted for $N = 0$, 1 and 2 with clear additional minima at $\nu_b = 1$, 3, 5, 7 and 9. Importantly, the $\nu = 0$ state that separates electron from hole states at $V_{bg}^{CNP}$ shows insulating behaviour with a diverging $R_L$, consistent with previous reports pointing to a gapped ground state[31–35]. At our highest $B$ (14 T), these broken symmetry states are furthermore accompanied by fractional QH

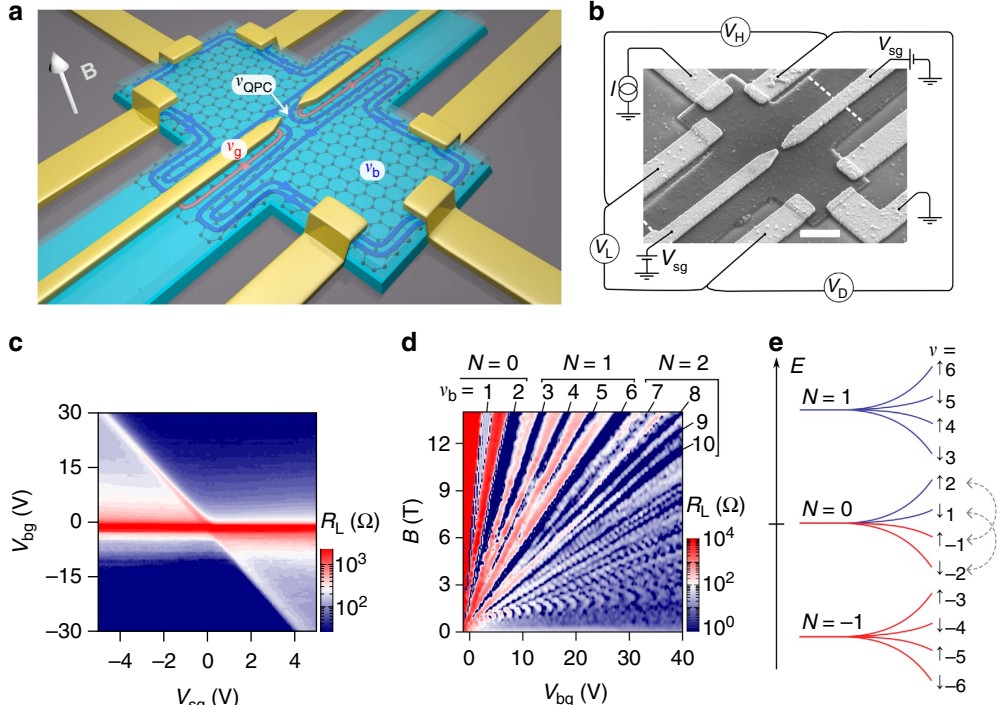

**Figure 1 | QPC device on hBN/graphene/hBN heterostructure.** (**a**) Schematic of the device showing graphene encapsulated in hBN (top hBN flake semitransparent) with electrodes contacting the graphene on the edge of the heterostructure. Edge channels formed at high magnetic field are shown as red (hole) and blue (electron) channels. $v_b$, $v_g$ and $v_{QPC}$ are the filling factors in the bulk, split gates and QPC regions, respectively, and determine the number and type of edge channels present. For this schematic, $v_b = 2$, $v_g = -1$ and $v_{QPC} = 1$. (**b**) Scanning electron micrograph of the device showing the measurement configurations. White dotted lines show the graphene edge buried below the hBN top layer. Scale bar is 1 μm. (**c**) Longitudinal resistance $R_L$ as a function of back-gate and split-gate voltages at zero magnetic field. (**d**) Landau fan diagram of longitudinal resistance measured at 0.05 K and $V_{sg} = 0$ V. Indexed blue strips indicate bulk QH states. (**e**) Energy diagram showing degeneracy lifting of the $N = -1$, 0 and 1 LLs into broken symmetry states indexed by the filling factor $v$. Arrows indicate the spin polarization of each electron (blue) and hole (red) level. Grey dashed arrows indicate the specific spin-selective equilibration restricted to the $N = 0$ LL.

plateaux that are pronounced in the Hall conductance (Supplementary Fig. 3).

**QPC operation in the integer QH regime.** Let us now investigate the control of integer QH edge channels by split-gate electrodes defining the QPC. We begin with a set of data taken at $B = 7$ T in the n-doped regime ($V_{bg} > V_{bg}^{CNP}$). Figure 2a displays the colour map of the diagonal conductance $G_D = 1/R_D$ across the split gates versus $V_{bg}$ and $V_{sg}$. For negative $V_{sg}$, conductance plateaux quantized in units of $e^2/h$ draw diagonal strips spanning a large range of $V_{bg}$, this voltage controlling the bulk filling factor $v_b$ (labelled on the right axis). At positive $V_{sg}$, these diagonal strips break up into rhombi that are horizontally delimited by the width of the bulk QH plateaux, centred at integer values of $v_b$. Diagonal grey dotted lines index the expected filling factor beneath the split gates, $v_g$, related to the local charge carrier density (extracted from Fig. 1c). These lines, together with $v_b$ labels, give the exact QH edge channel configuration beneath the split gates and in the bulk at any ($V_{bg}$; $V_{sg}$).

We first focus on the negative $V_{sg}$ regime where the charge carrier density below the split gates is lower than the graphene bulk density, as is required for confining bulk QH edge channels into the QPC. Inspecting the conductance strips we note that their slope does not match the lines of constant $v_g$. The shallower slope rather indicates a smaller capacitive coupling to the split-gate electrodes and hence a region of the graphene device with a charge carrier density in between those of the

bulk and the split gates. This region is nothing but the QPC saddle-point constriction that is capacitively coupled to both the back-gate and the split-gate electrodes. We thus introduce a third filling factor $v_{QPC}$ related to the charge carrier density in the QPC (see Fig. 1a). In the following, we demonstrate that this framework provides a fully consistent understanding of our data.

For negative $V_{sg}$, the filling factor configuration is $v_g < v_{QPC} < v_b$, where inner bulk edge channels are expected to be successively back-reflected at the QPC (see, for example, Fig. 1a). We first discuss the diagonal strips in yellow, red and dark red. Near zero split-gate voltage, charge carrier density is homogeneous in the whole graphene device, and $v_g = v_{QPC} = v_b$ with $v_b = 2$, 3 and 4, respectively. At these points the conductance values of $2e^2/h$, $3e^2/h$ and $4e^2/h$ are those of the respective bulk QH plateaux defined by $v_b$. The key feature of this series of strips is that their conductance remains constant for any higher $v_b$ and any lower $v_g$, indicating that only 2, 3 and 4 QH edge channels remain transmitted across the QPC, even if the number of edge channels increases in the bulk or decreases below the split gates. This clearly demonstrates that the strip conductance depends on the filling factor in the QPC and hence on the number of transmitted bulk edge channels. The diagonal conductance thus follows:

$$G_D = \frac{e^2}{h} v_{QPC} \qquad (1)$$

where $v_{QPC}$ counts the number of transmitted edge channels, as expected for standard QPC devices[6]. Consequently, we can index

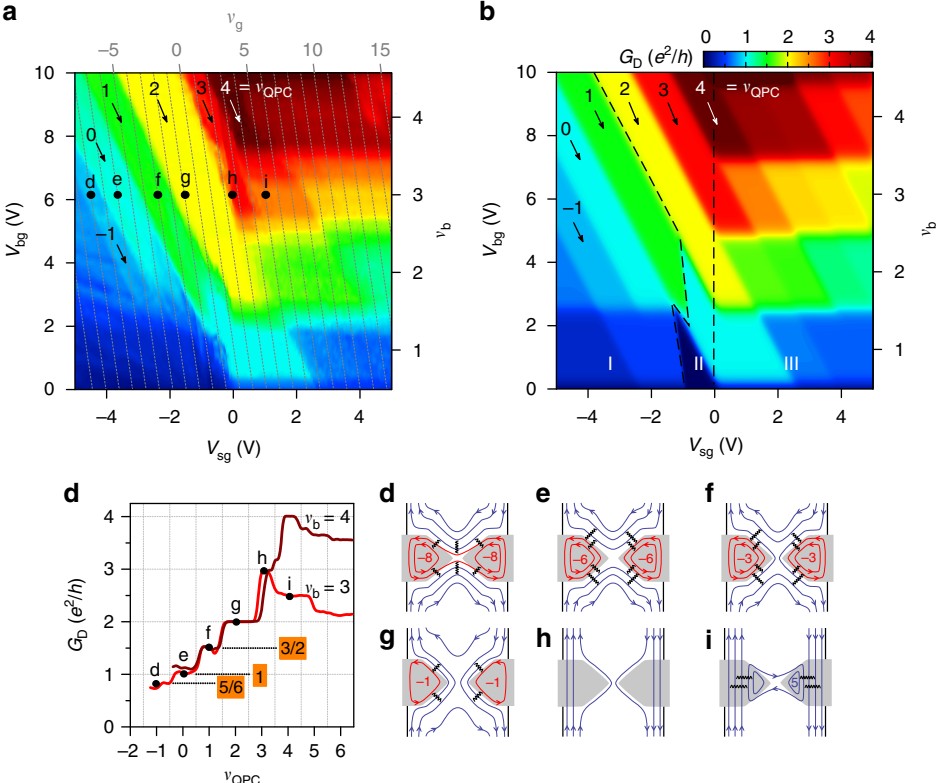

**Figure 2 | QPC in the quantum Hall regime.** (**a**) Diagonal conductance $G_D$ as a function of back-gate and split-gate voltages, $V_{bg}$ and $V_{sg}$, respectively. The bulk filling factor $\nu_b$ is labelled on the right axis. The grey dotted lines indicate constant filling factor below the split gates and are indexed by $\nu_g$ on the top axis. The diagonal arrows indicate constant QPC filling factor $\nu_{QPC}$. Note that the sample is current biased precluding measurement of vanishing conductance at full pinch-off. (**b**) Computed diagonal conductance map divided into three regions that delimit different operating regimes. In region I the QPC is short circuited by equilibration through the localized states beneath the split gates. Region II defines the QPC operating regime. Region III is analogue to n–n′–n top-gated structures. (**c**) $G_D$ versus $\nu_{QPC}$ for $\nu_b = 3$ and 4. The black dots labelled d–i correspond to the ones in (**a**). The labels 3/2, 1 and 5/6 highlight the anomalous conductance values due to equilibration in region I for $\nu_{QPC} = 1$, 0 and $-1$, respectively. (**d–i**) Edge channel configurations at the locations of the respective black dots in (**a**). Below the split-gates (grey areas), the numbers indicate $\nu_g$, with only the first two edge channels drawn. Equilibration between electron and hole channels is indicated by black wavy lines. (**d**) $(\nu_b, \nu_g, \nu_{QPC}) = (3, -8, -1)$, (**e**) $(3, -6, 0)$, (**f**) $(3, -3, 1)$, (**g**) $(3, -1, 2)$, (**h**) $(3, 2, 3)$ and (**i**) $(3, 5, 4)$.

$\nu_{QPC}$ according to (1) with the conductance value of each strip (the resulting values are labelled at the top of Fig. 2a).

This picture is further borne out by looking at the closing of the QPC with decreasing $V_{sg}$ from zero at fixed $V_{bg}$. Figure 2c shows line cuts of $G_D$ versus $\nu_{QPC}$ extracted from Fig. 2a at $\nu_b = 3$ and 4. The dark red curve taken at $\nu_b = 4$ exhibits a plateau of $G_D = 4\,e^2/h$ at $\nu_{QPC} = 4$ when four bulk QH edge channels are transmitted. Reducing $\nu_{QPC}$ to three by decreasing $V_{sg}$ enables only three edge channels to pass through the QPC, leading to a plateau of $G_D = 3\,e^2/h$, and similarly for $\nu_{QPC} = 2$ at lower $V_{sg}$. Therefore, closing the QPC by reducing the split-gate voltage leads to successive back scattering of the inner edge channels demonstrating QPC operation in the integer QH regime.

Upon further closing the QPC, the situation becomes more complex. Decreasing $V_{sg}$ to $\nu_{QPC} = 1$ does not result in a conductance of $e^2/h$, but $(3/2)\,e^2/h$ (green strip in Fig. 2a). Likewise, the conductance strips at $\nu_{QPC} = 0$ and $\nu_{QPC} = -1$ should show full pinch-off with $G_D \approx 0$, but instead we observe conductance plateaus at $\sim e^2/h$ and $\sim 0.85\,e^2/h$, respectively, for any $\nu_b \geq 2$.

The key to understanding these anomalous plateaux relies on a specific charge transfer—equilibration—between some of the back-reflected electron edge channels and some of the localized hole edge channels beneath the split gates, thus adding a new conduction path short circuiting the QPC. Following pioneering

works on p–n junctions in graphene[21–23] and assuming full equilibration, we solved the current conservation law for the QPC geometry that now involves three filling factors $\nu_b, \nu_g$ and $\nu_{QPC}$, thus complexifying equilibration compared with n–p–n junctions (see Supplementary Note 5). Taking into account that equilibration only occurs between QH edge channels of same spin polarization[24], the diagonal conductance reads:

$$G_D = \sum_{\sigma = \uparrow, \downarrow} G_D^{\nu_b^\sigma, \nu_g^\sigma, \nu_{QPC}^\sigma} \qquad (2)$$

$$G_D^{\nu_b^\sigma, \nu_g^\sigma, \nu_{QPC}^\sigma} = \frac{e^2}{h} \, |\nu_b^\sigma| \, \frac{2\,|\nu_b^\sigma||\nu_g^\sigma| + \nu_{QPC}^\sigma \left(|\nu_b^\sigma| - |\nu_g^\sigma|\right)}{3\,|\nu_b^\sigma||\nu_g^\sigma| + |\nu_b^\sigma|^2 - 2\,\nu_{QPC}^\sigma |\nu_g^\sigma|} \qquad (3)$$

where $\nu_b^\sigma$, $\nu_g^\sigma$ and $\nu_{QPC}^\sigma$ count the number of sub-LLs of identical spin polarization $\sigma$. Here $\nu_b^\sigma$ and $\nu_g^\sigma$ are of opposite signs, whereas $\nu_{QPC}^\sigma$ can be of both signs.

The fact that the conductance strips at $\nu_{QPC} = 1$, 0 and $-1$ become anomalous from $\nu_b = 2$ and stay constant for any higher $\nu_b$ and any $\nu_g < 0$ indicates that populating other LLs, namely $N \geq 1$ in the bulk and $N \leq -1$ under the split-gates, does not change the conductance. This finding thus points to equilibration uniquely between edge channels of the $N = 0$ LL. We therefore

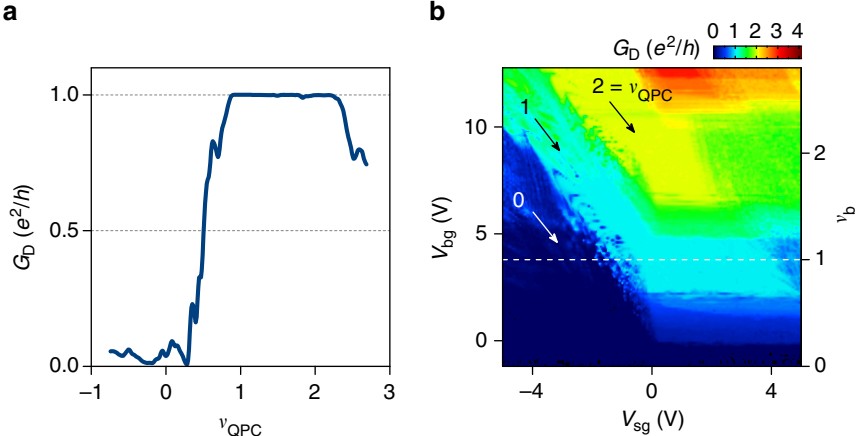

**Figure 3 | Full pinch-off.** (**a**) Diagonal conductance $G_D$ versus filling factor in the QPC $v_{QPC}$ demonstrating a conductance drop to zero and thus full pinch-off. The curve is extracted at the white dashed line in the diagonal conductance map shown in (**b**). (**b**) Diagonal conductance versus back-gate $V_{bg}$ and split-gate voltages $V_{sg}$ measured at 14 T and 0.05 K in voltage bias configuration (excitation voltage of 15 μV). The dark blue region corresponds to full pinch-off. The green strip at $v_{QPC} = 1$ appears at higher $v_b$ as compared with Fig. 2a, suggesting less effective equilibration at higher magnetic field.

infer that charge transfer is restricted to occur between the spin upward $v = 2$ and $v = -1$ sub-LLs, and between the spin downward $v = 1$ and $v = -2$ sub-LLs, as indicated by the dotted arrows in the LL energy diagram in Fig. 1e.

As a result, for $v_{QPC} = 1$ and $v_b \geq 2$, charge transfer between the back-reflected $v_b = 2$ edge channel and the localized hole $v_g = -1$ edge channel (see Fig. 2f) leads to $G_D^{1^\uparrow,-1^\uparrow,0^\uparrow} = (1/2) e^2/h$. As the $v_b = 1$ edge channel is transmitted through the QPC and contributes to $e^2/h$ to the conductance, the sum of both contributions gives $(3/2) e^2/h$, in agreement with the measured conductance value. For $v_{QPC} = 0$ and $v_b \geq 2$ as sketched in Fig. 2e, charge transfer between $v_b = 1$ and $v_g = -2$ edge channels (downward spin polarization), and between $v_b = 2$ and $v_g = -1$ edge channels (upward spin polarization), gives $G_D^{1^\downarrow,-1^\downarrow,0^\downarrow} = G_D^{1^\uparrow,-1^\uparrow,0^\uparrow} = (1/2) e^2/h$, thus a sum equal to $e^2/h$, as measured on the $v_{QPC} = 0$ strip. For $v_{QPC} = -1$, a hole state connects the split gates as sketched in Fig. 2d. In this case $G_D^{1^\uparrow,-1^\uparrow,0^\uparrow} = (1/2) e^2/h$ and $G_D^{1^\uparrow,-1^\uparrow,-1^\uparrow} = (1/3) e^2/h$, leading to a total conductance of $(5/6) e^2/h \simeq 0.83 e^2/h$, remarkably close to our measurement. Consequently, spin-selective equilibration restricted to the $N = 0$ LL provides a full explanation of the anomalous conductance values of the strips when the filling factor in the QPC is reduced to 1, 0 or $-1$.

If we now consider positive $V_{sg}$, the filling factor configuration changes to $v_b < v_{QPC} < v_g$. Extra electron-type QH edge channels can thus connect left and right edges of the graphene device (see Fig. 2i) leading to chemical potential equilibration as in n–n′–n top-gated structures[23]. We observe spin-selective partial equilibration in this configuration that is not restricted to the $N = 0$ Landau level, resulting in fractional conductance values similar to those reported in high-mobility graphene devices[24] (see Supplementary Note 5 for further analysis).

The above analysis of the QPC diagonal conductance is computed in Fig. 2b, taking into account spin-selective equilibration for both $V_{sg}$ polarities. Three distinct regions can be identified. In region I, equilibration restricted to the $N = 0$ LL between reflected electron edge channels and localized hole states short circuits the QPC leading to anomalous conductance plateaus. Region II defines the equilibration-free QPC device operating regime, where the conductance is precisely defined by the number of transmitted channels. Region III, at positive $V_{sg}$, describes the regime of n–n′–n unipolar equilibration. The remarkable one-to-one correspondence with our data of Fig. 2a supports the consistency of our analysis.

**Full pinch-off.** We complete this study of the QPC operation in the integer QH regime by addressing the pinch-off that occurs for $v_b = 1$ according to the dark blue triangle in region II in Fig. 2b. Figure 3a displays $G_D$ versus $v_{QPC}$ at $v_b = 1$ extracted from a set of data taken at 14 T (Fig. 3b). The conductance drops from $e^2/h$ to a value $< 0.1 e^2/h$, indicating full pinch-off of the $v_b = 1$ QH edge channel, when $v_{QPC} < 1$. Interestingly, we observed in the conductance map shown in the inset of Fig. 3 that $G_D$ remains vanishingly small (dark blue area) over a large range of gate values, indicating that equilibration is less efficient at 14 T than at 7 T, most likely due to the larger energy gaps.

**QPC operation in the fractional QH regime.** In the following, we turn to the QPC operation in the fractional QH regime. Owing to the high quality of our samples, fractional QH plateaux of the 1/3 family at $v_b \pm 1/3$ develop at a relatively low magnetic field of 14 T (refs 32,33,36,37). We observe plateaux in the bulk Hall conductance accompanied with minima in longitudinal resistance at bulk filling factors 1/3, 2/3, 4/3, 10/3 and 11/3. Other plateaux such as $v_b = 7/3$, 8/3 have, however, lower fidelity with the expected quantized values (see Supplementary Figs 3,4).

Akin to the integer QH effect, a key ingredient implicit in the analysis of transport properties in the fractional QH regime is the existence of fractional QH edge channels[38,39]. While their nature differs from the integer QH edge channels as they emerge from many-body interacting ground states within each LL[40], spatially separated edge channels are expected to propagate along the sample edges as evidenced by different means in GaAs[1]. In our graphene samples the QPC provides a perfect tool to unveil fractional edge channels by individually controlling their transmission.

Figure 4 displays $G_D$ versus $V_{sg}$ for $v_b = 4,3,2$ and a fractional bulk value of 2/3, all taken at 14 T. Solid lines indicate the region II of equilibration-free QPC operation. For $v_b = 4$ at $V_{sg} = 0$ conductance is that of the bulk: $4 e^2/h$. Upon reducing $V_{sg}$ the conductance decreases due to the back-reflection at the QPC of the inner $v_b = 4$ edge channel, and new intermediate fractional conductance plateaux emerge at $(11/3) e^2/h$ and $(10/3) e^2/h$, before reaching the integer plateau of $3 e^2/h$. Further pinching-off reveals plateaux at $(8/3) e^2/h$ and $(7/3) e^2/h$, though with less accurate conductance quantization. The 8/3 plateau is more pronounced for $v_b = 3$, whereas the 7/3 is absent. Similarly, on pinching-off $v_b = 2$, a clear intermediate plateau at 4/3 emerges.

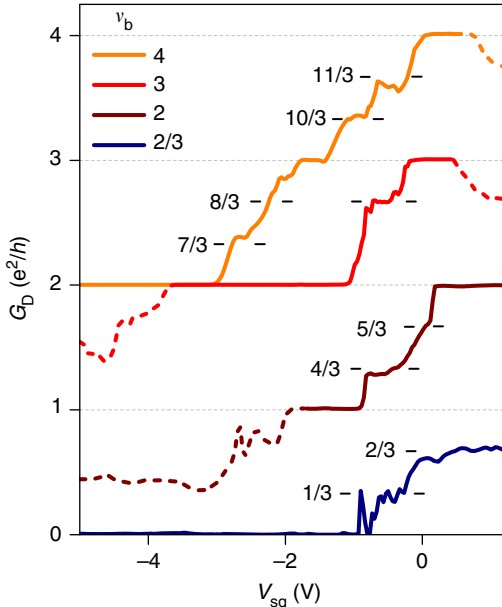

**Figure 4 | Pinch-off of fractional QH edge channels.** Diagonal conductance $G_D$ versus $V_{sg}$ for $v_b = 4$, 3, 2 and 2/3 measured at 14 T and 0.05 K. Solid lines indicate the regime II of QPC operation, whereas dashed lines relate to regimes I and III. Reducing split-gate voltage unveils intermediate plateaux at fractional conductance values that signal the successive back-reflection of fractional QH channels. The curves at $v_b = 4$, 3, 2 are measured in current bias configuration. The curve $v_b = 2/3$ is measured in 4-terminal voltage bias configuration, enabling measurement of the full pinch-off with $G_D \approx 0$ at $V_{sg} < -1$ V. The peaks in the transition to full pinch-off result from resonant tunnelling between counterpropagating edge channels of the 1/3 fractional state[52].

Note that the small kink at $G_D = (5/3) e^2/h$ signals the weak fractional state at $v_b = 5/3$, consistent with previous reports[36,37]. Generalizing equation (1) to the case of fractional filling factor $v_{QPC}$ (ref. 38), all these intermediate plateaux therefore unveil the successive back-reflection at the QPC of the respective fractional QH edge channels and thus their very existence. Eventually, upon increasing pinch-off while starting at $v_b = 2/3$, a clear $(1/3) e^2/h$ plateau emerges, followed ultimately, after its back-reflection, by a suppression of conductance indicating full QPC pinch-off.

## Discussion
In two-dimensional electron gases buried in semiconductor heterostructures, the nature of the charge transfer between edge channels has been investigated at length[4,5,41,42]. The overall picture is that any small amount of short-range disorder in real systems significantly enhances the tunnelling rate between adjacent channels[43–45]. In case of two co-propagating channels at different chemical potentials, this inter-channel tunnelling produces an out-of-equilibrium energy distribution that progressively relaxes to a new equilibrium by intra-channel inelastic processes. After complete equilibration, the current is equally distributed among channels that show identical chemical potential. In graphene, theoretical works showed that disorder and dephasing also drive equilibration at p–n junctions[46,47]. However, consideration of selection rules on spin or valley indexes for equilibration between broken symmetry states is still lacking.

Interestingly, in our high-mobility graphene devices, the regime of partial equilibration at $V_{sg} > 0$ shows inter-LL

equilibration, whereas in the QPC regime at $V_{sg} < 0$, equilibration is restricted to the $N = 0$ Landau level. This difference can be accounted for by the distinct paths taken by the bulk QH edge channels in the two regimes. At $V_{sg} > 0$, bulk edge channels indeed keep propagating along the graphene edges below the split gates, whereas for $V_{sg} < 0$ they are guided along the p–n junctions. These two paths markedly differ by the shape of the local electrostatic potential that is expected to be much smoother at the p–n junction than at the graphene edges. As the width of the incompressible strips that spatially separate edge channels is proportional to the ratio cyclotron gap over potential gradient[44], we expect the edge channels separation to be significantly increased at the p–n junction, especially between the $N = 0$ and $N = 1$ Landau levels that exhibit the largest cyclotron gap. As a result, at the p–n interface, the tunnelling rate between edge channels of different Landau levels should be exponentially suppressed due to the large incompressible strips, precluding inter-LL mixing as observed in the QPC regime (region I). Such conjectures that are based on electrostatics of QH edge channels call for further theoretical works including edge channel reconstruction and full degeneracy lifting of graphene QH states.

To conclude, our overall understanding of the precise edge channel configurations allows us to demonstrate gate-tunable and equilibration-free transmission of both integer and fractional QH edge channels through QPCs in graphene. Such a control of edge channel transmission enables future investigations of the equilibration processes at play that limit adiabatic transport[4,5,48], measurements of fractional charges[8,9] in the multicomponent fractional QH regime, design of single electron sources for electron quantum optics[12], QH interferometry[10,11] or even more prospective devices based on coupling QH states with superconducting electrodes[49–51]. Our work thus opens the way to a wealth of invaluable experiments in graphene exploring the variety of new QH ground states of both the integer[34,35] and fractional QH regimes[32,33,36,37].

**Data availability**. The data that support the findings of this study are available from the corresponding author on request.

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

## Acknowledgements

We are grateful to D. Abanin, H. Baranger, Th. Champel, H. Courtois, C. Dean, S. Florens, M. Goerbig, L. Levitov, Y. Meir and D. Shahar for invaluable discussions. This work was supported by the Nanosciences Foundation of Grenoble and the H2020 ERC Grant QUEST # 637815.

## Author contributions

K.Z. and B.S. conceived the experiment. K.Z., A.J. and Z.H. fabricated the samples; T.T. and K.W. grew hBN crystals used for the sample fabrication; F.G. provided technical support for the experiment; K.Z. and B.S. conducted the measurements; data analysis and interpretation were done by K.Z., H.S. and B.S.; the manuscript was written by B.S. with inputs from all authors.

## Additional information

**Competing interests:** The authors declare no competing financial interests.

**Publisher's note**: 

