## [Peer Review File · Nature Communications]

Reviewers' Comments:

Reviewer #1 (Remarks to the Author)

The paper by Zimmermann et al. reports on a study of edge channel transport through a quantum point contact on graphene in the quantum Hall effect. The authors perform their studies on devices of very high quality and obtain interesting results especially for the $N = 0$ Landau level. There are, however, a couple of issues that I would recommend to address carefully, before the paper can be considered for publication.

My impression is that the "storyline" of the paper needs to be revised. In order to differentiate their paper with respect to previous publications, the authors state that "recent experiments in graphene QPC devices operating on the fourfold degenerate Landau levels (LLs) showed that the presence of QH edge channel mixing is indeed detrimental as it creates a short circuit of the constriction via localized channels beneath the split-gates (red channels in Fig. 1a), thus hindering gate-control of QH edge channel transmission through the QPC [24, 25]." This is insofar problematic as most of the present manuscript discusses equilibration phenomena between co- and counter-propagating edge channels, as do Refs. 24 and 25. A statement like "Here we show that this electron-hole mixing is suppressed ..." in the abstract is not consistent with the sketches in Figs. 2 d-g. Furthermore, the authors claim in the abstract "QPC operation with full channel pinch-off", but they do not show pinch-off for the experimental situation of Fig. 2, because there is no region with $G_D = 0$ in Fig. 2a. The impression is that the authors oversell their results, but without necessity, because in my opinion, the real achievement of this paper are the experiments in the degeneracy-lifted lowest Landau levels, which are new and stimulating, and they were possible due to the much increased mobility of their samples with respect to previous studies ($200,000 \text{ cm}^2 / \text{Vs}$ instead of $\sim 15,000 \text{ cm}^2 / \text{Vs}$ in [24, 25]). I would therefore strongly recommend revising the manuscript in this sense.

The second issue, equally important, is that the authors do not discuss a physical mechanism for the proposed equilibration processes across channels. Due to the lifting of degeneracy, the equilibration is postulated to occur between channels which have another channel running in between them, see for example the sketch in Fig. 2g. This is a very interesting question, and it would be important to understand how this could happen (and I think for a publication in a Nature journal it would actually be required to provide a possible mechanism). I would expect that the intermediate channel screens the interaction between the channels to its left and right. Next, does the equilibration occur via exchange of quasiparticles (on page 8 the authors mention "charge transfer")? Or is there a capacitive coupling between the channels? If there is a transport of quasiparticles involved, I could understand why the quasiparticles do not jump into the neighboring channel, because spin or valley-index are wrong, but it should be considered that a scattering into the next-next neighboring channel is exponentially suppressed due to the larger distance between them. [This is briefly discussed in terms of the edge reconstruction model on page 11 of the Supplementary Information, but should be extended and included in the main text.] Also, for such a scattering process to happen, momentum needs to be supplied by some source. In a previous publication (Phys. Rev. B 83, 155305 (2011)) it has been shown that impurities can do this, but for samples as clean as the ones discussed here it would be questionable if there is a sufficient amount of defects in the vicinity of the QPC. If, on the other hand, a capacitive coupling is assumed, I do not see why this should be forbidden between neighboring channels, while allowed for next-next neighboring channels. To summarize this point, the authors should discuss a model for the proposed equilibration processes in the lowest Landau levels.

Reviewer #2 (Remarks to the Author)

Dear Editor,

the manuscript on „Tunable transmission of quantum Hall edge channels with full degeneracy lifting in split-gated graphene devices“ addresses an interesting and timely aspect of graphene research. Thanks to the increasing sample quality new experimental conditions are becoming available and the work by K. Zimmermann et al. is a good example for the nice quantum Hall physics that nowadays can be studied in graphene encapsulated by hBN. The authors demonstrate a (non-very surprising) tunable transmission of quantum Hall edge channels in a device geometry with split gates allowing for a kind of quantum point contact. The full pinch-off is in the reviewer's opinion the most interesting result of the presented work and it indeed opens a number of interesting avenues for further studies.

Unfortunately the authors discuss only one sample with partly limited data quality leaving some doubts on their interpretation. This holds in particular true for the discussion on the fractional quantum Hall edge state data (a single data set from a second sample is shown in the supplementary material, but only partly contributes to the discussion).

It is not clear to the reviewer why the authors do not show a complete high-resolution Landau level fan, similar to the one in figure 1d, including the hole side, and why they do not show close-ups of the transconductance to prove the (correct) slope of the fraction quantum Hall filling factors. In the reviewer's opinion these additional data are very important for unambiguously proving the presence of FQH states; figure S2 in the suppl. material is not sufficient for such a prove, in particular in view of the missing clear steps (and dips) in the shown traces. It would be necessary to show data as in figure S2 as function of B-field allowing to extract the individual slopes.

Overall, I would recommend to publish this work but only if some additional data (see points above) will be included in the manuscript.

Reviewer #3 (Remarks to the Author)

This manuscript describes the authors' experiment in quantum charge transport in a graphene device with split-gate in the quantum Hall regime. Particularly, they discussed the quantized charge transport through quantum point contact (QPC) with resolved four-fold degeneracy of Landau levels (LLs). From the obtained experimental data, they claimed that the spin-selective edge mode equilibration occurs under specific condition of top gate voltage. The proposed model, at a glance, explains the experimental data. However, it contains some fatal problems and inconsistency as follows.

1)The authors claimed that the spin-selective equilibrium and intermixing of edge channels appeared only within the quantized levels derived from the 0th LL under the condition of $V_{sg} < 0$. Why only negative V_{sg} ? The quantum Hall state is determined by the configuration of filling state which is controlled by both V_b and V_{sg} in this case. The authors should have discussed why such an unnatural assumption was raised.

2)Under the condition of positive V_{sg} , the edge channel mixing among levels including LLs of $N=1, -1$, etc., is allowed, while in the negative V_{sg} case it was restricted to only among edge channels originated from the $N=0$ LL. This is also unnatural and is hard to understand without any convincing explanation to justify the assumptions.

3)In the supplemental information, the authors referred to the spatial separation between edge channels. According to the description, the potential slope is steep at the edge of graphene flake, and so the spacing between edge channels become small. On the other hand in the pn junction, the potential slope is shallow, and so the spacing between edge channels become large which mitigates the mixing of edge channels. This contradicts the authors' explanation that the mixing of edge channels occurs at the pn junction. If the mixing of edge channels is enhanced at the edge of the graphene flake, equilibration among quantized levels from higher LL indices occurs under any

conditions and all of them must contribute to the quantum transport. This picture is also different from the authors' claims.

In conclusion, the proposed model of spin-selected mixing of edge channels is built with unnatural assumptions, and they are not based on appropriate physical background. The authors took some irrelevant assumptions unfairly to explain their experimental data, and their attempts resulted in inconsistent discussion. I do not recommend this manuscript for publication in Nature Communications.

Response to Reviewer #1:

We would like to thank Reviewer #1 for the valuable comments that helped us in improving our manuscript. Reviewer #1 specifically points out that i) our data on full QPC pinch off was not conclusive and that ii) a discussion on the equilibration process at play in our devices is missing. We reply below to these two aspects and present the ensuing modifications of the manuscript.

In order to differentiate their paper with respect to previous publications, the authors state that “recent experiments in graphene QPC devices operating on the fourfold degenerate Landau levels (LLs) showed that the presence of QH edge channel mixing is indeed detrimental as it creates a short circuit of the constriction via localized channels beneath the split-gates (red channels in Fig. 1a), thus hindering gate-control of QH edge channel transmission through the QPC [24, 25].” This is insofar problematic as most of the present manuscript discusses equilibration phenomena between co- and counter-propagating edge channels, as do Refs. 24 and 25.

Our work shows that, with broken symmetry states in high mobility devices, equilibration at the p-n junction is only restricted to the $N=0$ LL. This therefore enables an equilibration-free regime in some gate parameter range, where current is only carried by the transmitted edge channels, a situation that was precluded in previous works due to the fourfold degeneracy and the unavoidable equilibration. The “problematic” aspect of these sentences pinpointed by Reviewer #1 comes from the fact that the manuscript provides a clear and detailed description of the new equilibration regime that we uncover in our devices, which may sound as if equilibration properties was the main result of this work. However, we think that a detailed description of the different equilibration regimes remains important and deserves to be in the main text as it enables to define the exact regime of full (equilibration-free) QPC operation.

A statement like “Here we show that this electron-hole mixing is suppressed ...” in the abstract is not consistent with the sketches in Figs. 2 d-g.

We agree with the referee that this sentence may be misleading as it can be interpreted as a total absence of electron-hole mixing, which was not the message we wanted to convey. We have thus changed the sentence “Here we show that this electron-hole mixing is suppressed ...” to “Here we show that this electron-hole mixing is drastically reduced ...”, which is now correct as electron-hole mixing is restricted to the zeroth Landau level.

Furthermore, the authors claim in the abstract “QPC operation with full channel pinch-off”, but they do not show pinch-off for the experimental situation of Fig. 2, because there is no region with $G_D = 0$ in Fig. 2a.

The Reviewer is correct in pointing out that our statement of full pinch-off is not supported by Fig 2a and that we provide only one line-cut in Fig. 3 which is discussed in the paragraph on the fractional quantum Hall regime. As acknowledged by Reviewer #2, the full pinch-off is an important and decisive result for

the QPC operation and also for graphene physics. We provide below some clarifications on Fig2a and discuss the ensuing manuscript modifications.

Full channel pinch-off is limited to a narrow but well identified filling factor configuration. In Fig 2b, the region of full pinch-off is the small dark-blue triangle ($-1 < V_{sg} < 0$ and $0 < V_{bg} < 2$) beneath the number II. In the data of fig. 2a, there is indeed no region of $G_D=0$ in the same gate range. The reason for this is that measurements were performed in current-bias configuration, which cannot measure vanishing conductance (high resistance). We thus add in the caption of Fig2a: “*Note that the sample is current biased precluding measurement of vanishing conductance at full pinch-off*”.

Furthermore, we have revised the manuscript by adding a new figure 3 that displays a set of data taken at 14T in voltage-biased configuration. This figure shows clear pinch-off of the $\nu_b = 1$ bulk edge channel with a conductance $G_D < 0.1 e^2/h$ as well as the corresponding conductance map, which shows the expected dark blue region identified in Fig 2b. This figure is described in a new paragraph (“**Full pinch-off**. We complete this study...”) in page 10.

The second issue, equally important, is that the authors do not discuss a physical mechanism for the proposed equilibration processes across channels. Due to the lifting of degeneracy, the equilibration is postulated to occur between channels which have another channel running in between them, see for example the sketch in Fig. 2g. This is a very interesting question, and it would be important to understand how this could happen (and I think for a publication in a Nature journal it would actually be required to provide a possible mechanism). I would expect that the intermediate channel screens the interaction between the channels to its left and right. Next, does the equilibration occur via exchange of quasiparticles (on page 8 the authors mention “charge transfer”)? Or is there a capacitive coupling between the channels? If there is a transport of quasiparticles involved, I could understand why the quasiparticles do not jump into the neighboring channel, because spin or valley-index are wrong, but it should be considered that a scattering into the next-next neighboring channel is exponentially suppressed due to the larger distance between them. [This is briefly discussed in terms of the edge reconstruction model on page 11 of the Supplementary Information, but should be extended and included in the main text.] Also, for such a scattering process to happen, momentum needs to be supplied by some source. In a previous publication (Phys. Rev. B 83, 155305 (2011)) it has been shown that impurities can do this, but for samples as clean as the ones discussed here it would be questionable if there is a sufficient amount of defects in the vicinity of the QPC. If, on the other hand, a capacity coupling is assumed, I do not see why this should be forbidden between neighboring channels, while allowed for next-next neighboring channels. To summarize this point, the authors should discuss a model for the proposed equilibration processes in the lowest Landau levels.

The points raised by Reviewer, that is, the exact process involved in the equilibration, the role and nature of disorder, spin and/or valley-index selection rules, are all very interesting and would deserve a full research program for both theory and experiments (such as for instance the delicate and insightful scanning gate microscopy study mentioned by Reviewer). However, we want to recall that the scope of our work focuses on the demonstration of QPC operation in graphene in the QH regime. As such, our work, more precisely the QPC device geometry, is not aimed at providing explanation on the liable mechanism involved in the spin-selective equilibration occurring at a p-n junction in high mobility graphene.

Nevertheless, we fully agree with the referee that our first version of the manuscript was missing a discussion on equilibration (although we cannot provide new inputs on this issue with such specific QPC devices). We therefore add in the revised manuscript two paragraphs that introduce equilibration processes known in GaAs, and discuss the specific equilibration we observe. We indeed provide a possible explanation on the difference between the regime of partial equilibration ($V_{sg} > 0$) and QPC regime ($V_{sg} < 0$), where equilibration is restricted to the zeroth LL. This discussion, which was previously partially done in the supplementary materials, is now extended in the main text page 11-13.

Regarding the issue of spin-selection and the enhanced suppression of tunneling rate due to the hopping in second nearest neighbour channel, we do not think that our data and the spin-selective equilibration are inconsistent with a charge tunneling process. Tunneling rate will be indeed smaller between second nearest neighbour channels than between first neighbour channels. However, as with QH broken symmetry state the latter is forbidden, one cannot compare the efficiency of equilibration with and without spin-selection. Therefore, the spin-selective equilibration we observe can still result from disorder-induced tunneling, even if tunneling rate would have been much faster with first neighbour channels, if any. Note that the source of disorder is hard to determine but it could be due to edge imperfections of the gate electrodes that can lead to local potential fluctuations.

To summarize, we significantly modified the manuscript by adding a new figure and a paragraph demonstrating the full pinch-off of the QPC. We also added a discussion section on the issue of equilibration. We propose in this section a possible explanation in terms of enhanced width of the incompressible strips to explain the equilibration restricted to the zeroth Landau level.

Response to Reviewer #2

We are glad that Reviewer #2 emphasized that our work “addresses interesting and timely aspect of graphene research” and recommended publication after revision. Reviewer #2 has concerns about our fractional quantum Hall data. We therefore add to the manuscript a new figure, as suggested by Reviewer #2, which gives indeed a more convincing demonstration of the presence of FQH states in our devices. We thus would like to thank Reviewer #2 for his valuable comments and suggestions for a new figure, which helped us in strengthening our findings.

The manuscript on „Tunable transmission of quantum Hall edge channels with full degeneracy lifting in split-gated graphene devices“ addresses an interesting and timely aspect of graphene research. Thanks to the increasing sample quality new experimental conditions are becoming available and the work by K. Zimmermann et al. is a good example for the nice quantum Hall physics that nowadays can be studied in graphene encapsulated by hBN. The authors demonstrate a (non-very surprising) tunable transmission of quantum Hall edge channels in a device geometry with split gates allowing for a kind of quantum point contact. The full pinch-off is in the reviewer’s opinion the most interesting result of the presented work and it indeed opens a number of interesting avenues for further studies.

Thanks to Reviewer #1 comments, we extended the full pinch-off demonstration by adding a new figure 3 together with a new paragraph. With this new figure we now show full pinch-off of the QPC with a bulk filling factor equal to 1, in addition to the pinch-off of the fractional state $2/3$ presented in figure 4 (previously Fig. 3).

It is not clear to the reviewer why the authors do not show a complete high-resolution Landau level fan, similar to the one in figure 1d, including the hole side,...

We focused in this work on the electron side because contacts are more resistive on the hole side, leading to not well defined Landau level fan and Hall quantization (see figure A). Such an electron-hole asymmetry usually stems from a charge transfer from the metallic contact to the graphene. In our devices, contacts induce an electron-doped region around them in graphene, therefore forming a p-n junction when the bulk is hole-doped. In the quantum Hall regime, in line with the mitigated equilibration observed in the study of the QPC, the transmission of the contacts are reduced due to these p-n junctions, resulting in a bad contact transmission and thus a bad quantization and a noisy longitudinal resistance oscillations (See figure A). As the hole side of the data does not add value to the discussion and demonstration of the QPC operation, we do not think it is worth showing this data in the main text (if requested we can add those figures to the SI part).

...and why they do not show close-ups of the transconductance to prove the (correct) slope of the fraction quantum Hall filling factors. In the reviewers opinion these additional data are very important for unambiguously proving the presence of FQH states; figure S2 in the suppl. material is not sufficient for such a prove, in particular in view of the missing clear steps (and dips) in the shown traces. It would be necessary to show data as in figure S2 as function of B-field allowing to extract the individual slopes.

It is correct that dispersive features at fractional filling factor in the Landau fan diagram provides an excellent proof for the existence of FQH states. We thus added a new figure S3 to the supplementary information, which displays the derivative of the Hall conductance to V_{bg} plotted versus B and V_{bg} . This figure clearly shows minima at filling factors $2/3$, $4/3$ and $5/3$ dispersing with B with the correct slope. This provides a clear demonstration that the plateaus at fractional conductance values that we see both in the transverse conductance and in the QPC regime are fractional quantum Hall states. Note that, at our magnetic field of 14 T and within the resolution of the data, the minima in the longitudinal resistance were too faint to be visible in a Landau fan diagram colormap, and nearly impossible to visualize in a fan diagram due to the superimposed large variations of R_{xx} between integer filling factors, explaining why we have focused here on the Hall conductance.

Figure A | Left: Landau level fan diagram displaying $\log(\rho_{xx})$ versus B and V_{bg} . Right: colorplot of $\log(\rho_{xy})$ versus B and V_{bg} . Those are the same data as in Fig 1d of the main text.

Response to Reviewer #3

Reviewer #3 criticizes our finding of spin-selective equilibration restricted to the zeroth Landau level and considers it as an unnatural assumption. We reply below that our analysis relies on experimental facts and observations that can be accounted for by standard electrostatics of edge channels. Furthermore, we stress that Reviewer #3 does not dispute our main result, namely the demonstration of QPC operation both in the integer and fractional quantum Hall regime in a graphene van der Waals heterostructures.

1)The authors claimed that the spin-selective equilibrium and intermixing of edge channels appeared only within the quantized levels derived from the 0th LL under the condition of $V_{sg} < 0$. Why only negative V_{sg} ? The quantum Hall state is determined by the configuration of filling state which is controlled by both V_b and V_{sg} in this case. The authors should have discussed why such an unnatural assumption was raised.

2)Under the condition of positive V_{sg} , the edge channel mixing among levels including LLs of $N=1, -1$, etc., is allowed, while in the negative V_{sg} case it was restricted to only among edge channels originated from the $N=0$ LL. This is also unnatural and is hard to understand without any convincing explanation to justify the assumptions.

These two questions address the same issue. When hole states are present beneath the split gates at $\nu_g < 0$, we observe (not assume) that equilibration occurs between bulk electron channels and localized hole states, restricted to the $N=0$ LL. We want to emphasize that it is absolutely impossible to describe this regime at $\nu_g < 0$ by standard inter-Landau level equilibration as observed in other top-gated devices, or for partial equilibration when $\nu_g > 0$. Our statement on equilibration restricted to the zeroth LL at $\nu_g < 0$ results from observations in the data, as discussed at length in the paper. At any rate, our data – neither other published works on similar top-gated samples for the spin-selective equilibration— could provide a microscopic explanation on the exact origin of equilibration. Nevertheless, we added to the revised manuscript an extended discussion (previously partly in the SI) that provides an explanation in terms of width of incompressible strips that is expected to be wider at the p-n junction due to the smooth potential gradient, than at the graphene edge, thus suppressing the inter-Landau level edge channels tunneling rate. Our argument is only based on the standard electrostatic picture of QH edge channels.

3)In the supplemental information, the authors referred to the spatial separation between edge channels. According to the description, the potential slope is steep at the edge of graphene flake, and so the spacing between edge channels become small. On the other hand in the pn junction, the potential slope is shallow, and so the spacing between edge channels become large which mitigates the mixing of edge channels. This contradicts the authors' explanation that the mixing of edge channels occurs at the pn junction. If the mixing of edge channels is enhanced at the edge of the graphene flake, equilibration among quantized levels from higher LL indices occurs under any conditions and all of them must contribute to the quantum transport. This picture is also different from the authors' claims.

The new discussion added in page 11-13 better explains how the new regime of equilibration could be understood within the standard picture of edge channel reconstruction. The key point is the width of the

incompressible strips, which is proportional to the energy gap and inversely proportional to the potential gradient. As emphasized by Reviewer #1, the potential gradient should be shallow at the p-n junction. Therefore, the incompressible strips should be larger at the p-n junction than at the graphene edge. However, gaps between broken symmetry states are still quite small (about 1-10 meV) compared to the cyclotron gap between N=0 and N=1 LLs (of the order of 100meV). This leads us to the reasonable assumption that the spatial separation between edge channels at the p-n junction is mainly enhanced between the set of edge channels belonging to the N=0 and N=1 LLs, explaining why the tunneling rate is suppressed between these two sets, but not suppressed enough between the four broken symmetry states of the N=0 LL.

In conclusion, the proposed model of spin-selected mixing of edge channels is built with unnatural assumptions, and they are not based on appropriate physical background. The authors took some irrelevant assumptions unfairly to explain their experimental data, and their attempts resulted in inconsistent discussion.

We do not agree with the above Reviewer's assessment as all our analysis rely on thorough and clear observations of the data, which lead to a self-consistent demonstration. Our findings are purely based on facts and data, and can, in addition, be pictured consistently within the standard electrostatic theory of QH edge channel reconstruction. Nevertheless, it is clear that this new equilibration regime deserves further theoretical and experimental works that would go beyond the scope of our paper.

Reviewers' Comments:

Reviewer #1 (Remarks to the Author)

I have carefully read the various referee reports, the reply letter by the authors, and the revised versions of the manuscript and the Supplementary Information. Overall I have the impression that the authors have replied satisfactorily to all issues raised by the referees. For what concerns the second issue raised in my own referee report, I would be inclined to agree with the authors that this study probably goes beyond the scope of the present paper. I therefore recommend publication of the manuscript in its present form, also in the hope that these (excellent) data will stimulate further investigations on the detailed nature of the equilibration process in the $N=0$ Landau level.

Reviewer #2 (Remarks to the Author)

Dear editor,

please apologize my late reply. I have been reading carefully the revised manuscript. The authors addressed my previous concerns and I find the new supplementary figure S3 particularly helpful. The data quality presented in Figure 1d unfortunately remained the same and the total number of investigated samples too. I would strongly recommend to substitute the figure 1d by the complete LL-fan, which the authors included in the reply letter. This at least provides more details on the quality and performance of the sample investigated.

Overall, the data quality could be higher but the main claims are supported by the presented measurements. Thus, I would agree recommending to publish this work.

Reviewer #3 (Remarks to the Author)

Following the authors' response and the revised manuscript, I realized that I was confused and misunderstood the configurations. Now, my concerns with the first two points have been resolved. And for the third point on the intermixing of edge channels, the added arguments are fair and should arouse further fruitful discussions on the equilibrium mechanism.